# Comparison of Epidemiological Data of Complex Regional Pain Syndrome (CRPS) Patients in Relation to Disease Severity—A Retrospective Single-Center Study

**DOI:** 10.3390/ijerph20020946

**Published:** 2023-01-04

**Authors:** Julian Diepold, Christian Deininger, Berndt-Christian Von Amelunxen, Amelie Deluca, Paul Siegert, Thomas Freude, Florian Wichlas

**Affiliations:** 1Department of Orthopedics and Traumatology, University Hospital Salzburg, Müllner Hauptstraße 48, 5020 Salzburg, Austria; 2Institute of Tendon and Bone Regeneration, Spinal Cord Injury & Tissue Regeneration Center Salzburg, Paracelsus Medical University, Strubergasse 21, 5020 Salzburg, Austria; 31st Orthopaedic Department, Orthopaedic Hospital Speising, Speisinger Straße 109, 1130 Wien, Austria

**Keywords:** CRPS, retrospective data analysis, Budapest criteria, wrist, radius fractures

## Abstract

A retrospective data analysis of 159 complex regional pain syndrome (CRPS) patients (n = 116 women, 73.0%, mean age 60.9 ± 14.4 years; n = 43 men, 27.0%, mean age 52.3 ± 16.7 years) was performed from 2009 to 2020. The right side was affected in 74 patients (46.5%), the left in 84 patients (52.8%), and 1 patient (0.7%) developed a bilateral CRPS. Data were analyzed for the frequency and distribution of symptoms. The number of reduction maneuvers and the number of Budapest criteria were compared in relation to the severity of CRPS. Hand and wrist (n = 107, 67.3%), followed by foot and ankle (n = 36, 22.6%) and other locations (n = 16, 10.1%) were mainly affected by CRPS. The main causes included direct trauma (n = 120, 75.5%), surgery without previous trauma (n = 25, 15.7%), other causes (n = 9, 5.7%), and spontaneous development (n = 3, 1.9%); there was also missing documentation (n = 2, 1.3%). The most common symptoms were difference in temperature (n = 156, 98.1%), limitation of movement (n = 149, 93.7%), and swelling (n = 146, 91.8%). There was no correlation between the number of reduction maneuvers and the number of Budapest criteria. In summary, patients with the following constellation are at increased risk of CRPS: a female, over 60 years old, who has fallen and has sustained a fracture in the hand or wrist with persistent pain and has been immobilized with a cast for approx. 4 weeks.

## 1. Introduction

Complex regional pain syndrome (CRPS) is a neurological, orthopedic, traumatological disorder. It is a chronic pain condition of poorly understood origin that predominantly affects distal parts of one extremity. The term CRPS was established internationally in recent years, but its origin and occasion are still not completely clear as of now [1,2].

The estimated overall incidence of CRPS, according to the literature, is 26.2 per 100,000 person years [3]. Of these, approximately 2 in 1000 patients who recover from a distal radius fracture develop CRPS [4]. 

CRPS mainly manifests in elderly females who experience a fracture of an upper extremity (UE) [5]. Prolonged immobilization, the number of reduction maneuvers, and excessively tight casts also increase the rate of CRPS [6,7]. Psychosocial circumstances, such as drastic life changes and a patient’s psychological state, also seem to influence its development. The pathophysiology underlying the development of CRPS is still being explored, but the disorder is believed to result from central and peripheral nervous system dysfunction [8,9].

### Objectives

Due to the complexity of CRPS, its treatment warrants the involvement of an interdisciplinary team including trauma/orthopedic surgeons, anesthesiologists, physiotherapists, and psychotherapists. Early diagnosis and initiation of treatment is therefore critical to limit disease progression and improve patients’ quality of life.

Nowadays, the disease is mainly verified clinically by the occurrence of a characteristic constellation of symptoms and includes excruciating pain which cannot be explained by the fracture itself [10]. The occurrence of other symptoms such as swelling, skin discoloration, and sweating also plays a role. In order to simplify its diagnosis, the Budapest criteria were introduced [11,12,13]. Overall, a clinical examination remains the gold standard for diagnosing CRPS. The diagnosis of CRPS after distal radius fractures remains largely clinically based on symptoms and physical examination [14], and that clinical examination remains the gold standard for the diagnosis of CRPS. 

Thus, the aim of this study is to analyze the epidemiological and clinical data of CRPS patients to identify further specific factors that promote its development. In particular, patients with a special constellation of symptoms (Budapest criteria) and demographic characteristics (e.g., age and gender) should be filtered out in order to prevent the possible development of CRPS. We hypothesize that by minimizing those promoting factors, the overall risk of developing CRPS can be reduced.

## 2. Materials and Methods

From 2009 to 2020, 159 patients with CRPS were analyzed retrospectively in a level I trauma center (University Hospital Salzburg, Salzburg, Austria). Data were selected from the hospital’s own thermography device (TVS-500EX Thermal Imaging Camera, Nippon Avionics Co., Ltd. 224-0053, Yokohama, Japan) and digital patient system software (ORBIS, Dedalus Health Care, 53227 Bonn, Germany). 

Epidemiological and demographic parameters including gender, age, the affected side and extremity (upper extremity (UE) and lower extremity (LE)), and the anatomic region were evaluated. Further, symptoms of patients with CRPS, treatment modalities, and the number of Budapest criteria were included in the study. 

Possible trigger events for CRPS development were analyzed, such as the occurrence of CRPS after trauma, operative versus non-operative treatment modalities, occurrence after elective surgery, and cases where no trigger event could be detected. In CRPS patients with distal radius fractures, the number of reduction maneuvers, the duration of cast immobilization, and the type of operation were recorded.

To confirm the diagnosis, the revised Budapest criteria of the IASP (International Association for the Study of Pain) were used. [15]. At least six symptoms in the patients’ medical history and the clinical examination must be present to confirm the presence of CRPS.

### 2.1. Budapest Clinical Diagnostic Criteria for CRPS

All patients included in this study fulfilled the Budapest criteria including:Persistent pain, which is disproportionate to the initial trauma.Patients must report at least one symptom in four of the following categories:Sensory—hyperaesthesia and/or allodynia.Vasomotor—temperature asymmetry and/or skin color changes and/or skin color asymmetry.Sudomotor/oedema—oedema and/or sweating changes and/or sweating asymmetry.Motor/trophic—decreased range of motion and/or motor dysfunction (weakness, tremor, or dystonia) and/or trophic changes (hair, nail, or skin).
Patients must display at least one sign at the time of clinical evaluation in two or more of the following categories:Sensory—hyperalgesia and/or allodynia.Vasomotor—temperature asymmetry (>1 °C) and/or skin color changes and/or asymmetry.Sudomotor/oedema—oedema and/or sweating changes and/or sweating asymmetry.Motor/trophic—decreased range of motion and/or motor dysfunction (weakness or tremor) and/or trophic changes (hair or nail).
There is no other diagnosis that better explains the signs and symptoms [16].

To confirm the diagnosis under the Budapest criteria, a person must have at least one symptom in three of the four categories and at least one sign must be present in two or more of the categories. In addition, no other diagnosis would explain the symptoms and signs better [17].

### 2.2. Statistical Analysis

Continuous variables were expressed as mean ± standard deviation (SD) and categorical variables as percentages (%). Differences for categorical variables were assessed with the chi-square test and considered statistically significant if the null hypothesis could be rejected with >95% confidence (*p* < 0.05). For pairwise comparisons, an unpaired *t*-test or Mann–Whitney test was used. A multivariable logistic regression analysis was done using age, extremity, difference in temperature, and trauma and surgery as predictors. Odds ratios with 95% CI were computed. All tests were performed using GraphPad Prism v. 9.02 (San Diego, CA, USA).

## 3. Results

### 3.1. Study Population

A total of 116 women (73.0%) with an average age of 60.9 ± 14.4 years and 43 men (27.0%) with an average age of 52.3 ± 16.7 years were included. The total average age of all participants was 58.5 ± 15.5 years. The right side was affected in 74 patients (46.5%), the left in 84 patients (52.8%), and one patient had bilateral limb involvement (0.7%). The UE was affected in 118 patients (74.2%) and the LE in 41 patients (25.8%). In women, the UE was affected in 86 cases (74.1%) and the LE in 30 cases (25.9%). In men, the UE was affected in 32 cases (74.4%) and the LE in 11 cases (25.6%).

### 3.2. Anatomical Region

In 107 cases (90.7%), the hand or wrist was affected: in 34 (31.8%), the hand (phalangeal, metacarpal, or carpal bones), in 68 (63.6%), the wrist, and in 5 (4.7%), not specified in more detail. In the remaining 11 (9.3%), the elbow, shoulder, or acromioclavicular (AC) joint were involved. 

Regions of the LE included the foot and ankle in 36 (87.8%) patients, and in the 5 remaining cases (12.2%), the tibial shaft, knee, or patella were affected. According to the statistical analysis of our dataset, the involvement of the UE was statistically more significant compared to the LE (*p* < 0.00001).

### 3.3. Trigger Events

In 120 (75.5%) patients, the trigger event was an unspecific trauma (e.g., a stumbling fall) which was treated either operatively (n = 69, 57.5%) or conservatively (n = 51, 42.5%). In 34 (21.4%) cases, the trigger event was related to medical treatment not related to trauma, of which 25 (73.5%) cases were treated operatively and 9 (26.5%) conservatively. In five (3.1%) cases, the source was unknown (Table 1).

### 3.4. Reduction Maneuvers of Fractures at the Wrist

In total, there were 53 fractures of the wrist. Of the 50 documented wrist fractures, all were reduced pre-operatively by closed reduction. Of these, 19 (65.5%) fractures were preoperatively reduced a single time and 10 (34.5%) more than once. 

Fifteen (71.4%) of the conservatively treated wrist fractures were reduced once by closed reduction. One patient’s fracture was reduced twice (5.0%), two (9.5%) fractures were not reduced at all, and in three cases, documentation was incomplete (Table 2). 

Regarding the lower extremity, seven closed reduction attempts were made. Of the ankle fractures, four (57.1%) were reduced one time, one (14.3%) twice, and two (28.6%) were not reduced.

### 3.5. Immobilization in Relation to the Entire Collective

A total of 64 (40.3%) patients were immobilized by a cast after surgical treatment and 44 (27.7%) patients during conservative treatment. No documentation was available in nine (5.7%) cases. The duration of cast immobilization for operatively and non-operatively treated patients are summarized in Table 3. 

In the operative group, 28 patients (30.4%) were treated without a cast after surgical intervention. This included patients who received an additional soft tissue surgery, such as a carpal tunnel splitting in 11 patients (39.0%). Of the operated patients, 26 (40.6%) received a cast immobilization for more than 6 weeks. This included mainly patients with a distal radius fracture. 

In the non-operative group, 16 patients (36.4%) were immobilized with a cast for more than 6 weeks and exclusively included patients with fractures. In total, 14 (24.1%) patients were not immobilized, did not suffer from any fractures, but did suffer from distortions.

### 3.6. Budapest Criteria

The number of symptoms each individual patient presented with at the time of diagnosis were collected from the hospital’s own digital patient archive software (ORBIS). The distribution of Budapest criteria ranged from a minimum of three to a maximum of thirteen symptoms. To confirm the diagnosis under the Budapest criteria, a person must have at least one symptom in three of the four categories and at least one sign must be present in two or more of the categories. In addition, no other diagnosis would explain the symptoms and signs better [16]. Most patients experienced seven symptoms (n = 69, 43.4%) according to the Budapest criteria. Of these, 69 were women (n = 51, 73.9%) with mainly a wrist pathology (n = 30, 43.5%). A total of 15 patients had fewer than 6 symptoms at initial contact. However, in those patients, further symptoms developed over the course of disease progression.

The distribution of the Budapest criteria is summarized in Table 4.

The most common documented symptoms, in addition to pain which could no longer be explained by the initial trauma, were asymmetric temperature (>1–2°), reduced range of motion (ROM), and edema of the affected region. Changes in hair and nail growth, hyperalgesia, and asymmetric sweating were less common.

In 123 patients (77%), the affected side was hotter than the contralateral one, and in 26 patients (16%), it was colder. In four patients (3%), there was no temperature difference, and in six (4%), the documentation was missing. Comparing the amount of temperature difference with the number of Budapest criteria shows that the higher the temperature difference, the higher the Budapest criteria number was. This corresponded only to the warmer affected extremity. On average, the affected side was 1.3 °C warmer than the unaffected side.

Age (*p* = 0.005), extremity (*p* = 0.009), and the interaction between age and extremity (*p* = 0.025) were found to be significant. A more detailed analysis in the subgroups revealed that age was found to be significant in the lower extremity patient group but not in the upper (odds ratio for age of 1.09 (95% CI: 1.027–1.16)). A figure for illustration is given in Figure 1.

## 4. Discussion

The origin and occurrences of CRPS are still being discussed controversially in the literature. In terms of gender, region, and of the disease cause, the context of the study population plays a key role. Pak et al. carried out a study in a hospital setting which confirmed a high occurrence of CRPS in females and is concurrent with our study results [17]. Other studies, including young Korean soldiers, differ mainly in the affected body region with a higher incidence of CRPS in the LEs and are mainly due to sprain or strain [18].

Controversially, the results of our study confirm that the wrist and hand are most likely to be affected by CRPS, whereas the second most common affected anatomic region is the ankle and foot. 

It is well known that a prolonged cast immobilization might enhance the development of CRPS [7]. Unfortunately, many patients in our study with a distal radius fracture also required a prolonged duration of postoperative immobilization and several cast changes, which might have increased the risk of developing CRPS. 

The duration of cast immobilization has been shown to be a crucial factor for the development of CRPS. In our study cohort, 58.5% of all patients had a cast for 4 weeks or longer. The amount of postoperative immobilization is higher compared to other studies, especially because ‘postoperative immobilization’ is a major factor contributing to the development of CRPS [14,19]. Allen et al. described a cast immobilization with a mean duration of 3 weeks in 47% of all patients [20]. Maves and Smith carried out studies with rodents which further confirmed a clear correlation between immobilization and the development of CRPS [21]. Nowadays, modern implants permit early physiotherapy and surgical aftercare without additional immobilization which seems to be crucial in preventing CRPS [22]. This again has been confirmed in an experimental setting: additional immobilization after internal fixation has been shown to be counterproductive as it combines the disadvantages of both treatment modalities [23]. Either you operate and move early, or you treat it conservatively with cast immobilization. 

Nevertheless, osteoporotic fractures often require additional external fixation (e.g., cast immobilization) in addition to the internal fixation because the bone–implant interface is weak. If those fractures are severely dislocated, surgery is required for adequate reduction, and due to the reduced bone quality, an additional external cast should prevent secondary implant failure. In our collective, the mean patients’ age of the surgically treated distal radius fractures was 69 ± 12.1 years and classified as geriatric. With increasing age, the incidence of developing dementia is on the rise and makes the post-surgical instructions to be followed sometimes difficult for this patient cohort. Many patients are unable to adhere to strict unloading and weight bearing of the affected limb and therefore require additional cast immobilization. This could explain the high number of patients treated with an additional cast. 

However, initial trauma preceded in 75.5% of cases the development of a CRPS, while 15.7% occurred after elective surgery. While trauma is an uncontrollable event in terms of its occurrence and course, elective surgery is well planned and allows the elimination of CRPS causing factors such as pain and long cast immobilizations.

The distribution of the Budapest criteria amongst various groups (trauma operative, trauma non-operative, and operation without trauma) was similar. The three most common criteria, besides pain, included the difference in temperature, the reduced ROM, and edema. The difference in temperature has been discussed controversially in the literature but seems to be a valid method of confirming the diagnosis [24]. In our collective, most patients had a higher temperature in the CRPS affected extremity compared to the unaffected side. The reduced ROM and edema were typical postoperative and posttraumatic signs which made them difficult to differentiate from a developing CRPS. Only 26.4% presented with an accelerated growth of fingernails and hair. 

Eleven distal radius fractures were reduced more than once, ten were treated operatively, and one non-operatively. Comparing the once-reduced distal radius fractures to the multiple-reduced ones in terms of the number of symptoms, there was no difference in the number of symptoms in relation to the number of reduction maneuvers in our patient collective. Thus, according to our data, multiple reduction attempts cannot be counted as a risk factor for developing CRPS. This is also in agreement with the study by Atkins et al. and Dijkstra et al. [25,26]. However, it is controversial to some other literature findings from the past: closed reduction of distal radius fractures was considered to be a potential risk factor for developing CRPS [6].

These findings are unexpected as closed reduction of distal radius fractures was considered to be a potential risk factor for the development of CRPS, especially repeated reduction maneuvers [6]. Many authors recommend carrying out one reduction maneuver only and to initiate an operative treatment if the obtained reduction is not satisfactory. Nevertheless, modern concepts for the treatment of dislocated wrist fractures include open reduction after a single reduction attempt and early mobilization of the wrist after surgical treatment.

A limitation of this study is the incomplete documentation of the medical history and the presented symptoms and, as a consequence, the documentation of the Budapest criteria. Since the patients were treated by different doctors, physiotherapists, ergo therapists, and physical doctors, no standardized documentation was maintained. Furthermore, not all patients were treated at the hospital from the beginning. Some patients were referred externally when the disease was already present.

## 5. Conclusions

In summary, when the following five factors coincide, the likelihood of CRPS development is significantly higher if: an (1) elderly lady (2) approx. 60 years of age with persistent pain (3) has fallen and has sustained a fracture in the (4) hand or wrist area and has been immobilized in a (5) cast for approx. four weeks. Especially when the lower extremity is affected, the severity of CRPS increases with age.

## Figures and Tables

**Figure 1 ijerph-20-00946-f001:**
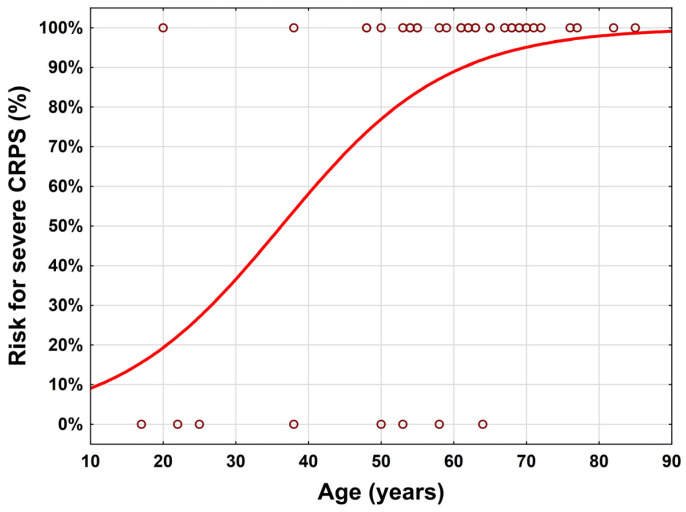
Risk for severe CRPS depending on age and lower extremity.

**Table 1 ijerph-20-00946-t001:** Summary of the occurrence of CRPS.

Source of CRPS (n = 159)					
Trauma (n = 120, 75.5%)		No Trauma (n = 34, 21.4%)		Other (n = 5, 3.1%)	
Surgery	Non-Operative Treatment	Surgery	Other	Idiopathic	Missing Documentation
n = 69 (57.5%)	n = 51 (42.5%)	n = 25 (73.5%)	n = 9 (26.5%)	n = 3 (60%)	n = 2 (40%)

**Table 2 ijerph-20-00946-t002:** Reduction maneuvers of fractures at the wrist (in three cases, documentation was incomplete).

Amount of Closed Reductions	0	1	2	3	4	Total
**Operative** **Treatment**	0 (0%)	19 (65.5%)	6 (20.7%)	3 (10.3%)	1 (3.4%)	29 (100%)
**Non-operative Treatment**	2 (9.5%)	15 (71.4%)	1 (5%)	0 (0%)	0 (0%)	21 (100%)
**Incomplete** **Documentation**						3 (100%)

**Table 3 ijerph-20-00946-t003:** Duration of cast immobilization for operative treatment and non-operative treatment.

Patients with Cast Immobilization (n = 108)										
**Operative Treatment (n = 64)**										
**Cast Immobilization (weeks)**	**11**	**9**	**8**	**7**	**6**	**5**	**4**	**3**	2	1
**Number of Patients** **n (%)**	0 (0.0)	1 (1.6)	1 (1.6)	1 (1.6)	23 (35.9)	11 (17.2)	16 (25.0)	4 (6.3)	5 (7.8)	2 (3.1)
**Non-operative Treatment (n = 44)**										
**Cast Immobilization (weeks)**	11	9	8	7	6	5	4	3	2	1
**Number of Patients (n) (%)**	1 (2.3)	1 (2.3)	0 (0.0)	2 (4.5)	12 (27.3)	17 (38.6)	7 (15.9)	2 (4.5)	0 (0.0)	2 (4.5)

**Table 4 ijerph-20-00946-t004:** The distribution of the Budapest criteria.

Budapest Criteria	3	5	6	7	8	9	11	13
**Patients (n)**	2	13	1	69	1	45	26	2
**Percent (%)**	1.3	8.2	0.6	43.3	0.6	28.3	16.4	1.3

## Data Availability

The data presented in this study are openly available in [repository name e.g., FigShare] at [doi], reference number [reference number].

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
