# Peer review of "Comparison of Epidemiological Data of Complex Regional Pain Syndrome (CRPS) Patients in Relation to Disease Severity—A Retrospective Single-Center Study"

_ijerph, 2023, doi:10.3390/ijerph20020946_

Round 1
Reviewer 1 Report
Dear Authors:
- Within section 1. INTRODUCTION, at the end of it, the objective is to identify the factors that favor the development of the CPRS, but in the previous paragraph we already talk about the precipitating factors.
- Within the material and methods section, the Budapest criteria should be reflected (they are not included in any table or mentioned as such in any part of the article), since it is a diagnostic material used and is later explained in lines 67 and 68 how to confirm CRPS.
- On page 2, in results, line 86 does not reflect what the acronyms AC mean (these acronyms are not appreciated with their meaning before the document).
- In line 91, what is a non-specific trauma? Could you give an example?
- In line 101 it must be specified, for a good understanding of the paragraph, that it begins to speak of fractures that did not require surgical treatment.
- In section 3.4. the general results of wrist fractures are discussed and then distal radius fractures are specified, as if it were independent. It should be clear which are the wrist fractures (if they are of the distal radius, carpal bones, etc.). Later he goes on to talk about ankle fractures. What happens with the rest of the fractures? Those of the elbow, shoulder, tibia, patella, etc... A table similar to Table 2 could be made, reflecting all the fractures studied.
- Line 116 shows that there are 14 people who did not receive immobilization, but table 3 shows that there are 12. You have to know if it is an error or if 2 patients are missing.
- In section 3.5. One should talk about fractures of the foot, ankle, breast or wrist that did or did not receive immobilization and for how long, since, according to the article, they are the most affected.
- In table 3, in the row of immobilization with plaster, it starts from 0 weeks to 11, without there being 10. Is it an error? And where should I put 11 is 10? o No one was immobilized for 10 weeks in both surgical and non-surgical treatment.
- A section of objectives is missing. It should not be included within the introduction.
- Bibliographical references number 1 and 17 are incomplete.
Author Response
Dear Reviewer,
thank you for rising the important points. In the following document we have implemented all suggestions for improvement.

Reviewer 2 Report
Line 2: the title should go without an acronym (CRPS) or first the words and then the acronym in parentheses.
Complex regional pain syndrome (CRPS)
Line 15: The mean age 58.5+/-15.5 of introduction is the mean age of all patients, not that of women. What is the mean age of men?
Are there differences between the ages of men and women?
Line 20: add other locations 16 (%)
Line 40: the citations about the psychosocial state of the patients are from the last century, I suppose there will be more recent citations.
Line 79: Initially describe the total number of patients: 159, of them 116 women and 43 men...
Line 79: what was the mean age of the men.
Are there differences between men and women?
Line 83: What is the distribution between men and women?
Are there differences in the distribution between men and women?
Line 87: add the total number 36 (87.8%)
Line 98: What is the total number of wrist fractures? 50 or 29 patients?
According to Table 2, the total would be 50 patients: 29 with operative Treatment, 18 with Non-operative Treatment and 3 unknown.
Line 98: previously 107 cases of hand and wrist involvement are cited (line 85), now 29 wrist cases are being analyzed, what has happened to the others?
Line 99: The percentage of 10 is 34.5% not 35.5%.
Line 99: 15 (71.4%) of the radius fractures... it did not appear before how many radius fractures there were in the study, according to the percentages it would be 21, would they be part of the 29 of the wrist or are they different?
Line 102: add the rest of EESS fractures are 7 with the following distribution...
Table 2: add a column with "incomplete data"
Line 106: according to the table of the 92 patients with operative treatment, 28 were not immobilized (30.4%), therefore the analyzes should be done with 64 patients
Line 107: according to the table of the 58 patients with non-operative treatment, 12 were not immobilized, so the analysis should be carried out with 46 patients.
Line 112: it would be better to give the data on patients with immobilization for more than 6 weeks, which were 23+1+1+1= 26 patients and give the percentaje of that data.
Line 115: Why is the figure of 5 weeks given here and not 6 as in the previous paragraph? It would be nice to give the figure of more than 6 weeks, as in the previous one.
Table 3: the sum of patients with non-operative treatment is 56, not 58.
Line 172: would it be possible to know how many of the patients analyzed in the study had osteoporosis criteria?
Are there differences between those who were operated or not?
General considerations:
1. Are there differences between men and women?
2. Are there differences between EESS and EEII?
3. It is advisable to check a large part of the results and then make the conclusions and the discussion, since it is carried out on data that is not correct.
4. Regarding the bibliography, 14 of 21 citations are more than 20 years-old and 3 of 21 are less than 5 years-old. It would be nice to have more recent comparative data, if possible.
Author Response

(The authors gave the same response as above.)

Reviewer 3 Report
"Comparison of epidemiological data of CRPS patients in rela- 2 tion to disease severity - a retrospective single-center study" is an interesting study.
Advices:
- Introduction and methods have to be extended.
- Methods has to be comlpettly edited - e.g. participants; Budapest scale.
- Literature needs edition - newer ones.
- Language has to be be improved (checked by a native speaker)
Author Response

(The authors gave the same response as above.)

Reviewer 4 Report
This study aim to investigate epidemiological and clinical risk factor of complex regional pain syndrome (CRPS). This is an interesting study and had some new information on it. But there is still some queries need to be confirmed first before we can accept it.
1. there is limited rationality about this study. authors need to add another paragraph about recent study regarding CRPS.
2. please explain it in detail about Budapest criteria because it is one of the main diagnosis tools for CRPS.
3. Please re-analyse the data by using multivariate analysis not only bivariate analysis. because bivariate analysis is not enough and had low internal validity. the conclusion could not be made if authors didn't do multivariate analysis.
Author Response

(The authors gave the same response as above.)

Round 2
Reviewer 3 Report
"Comparison of epidemiological data of CRPS patients in rela-tion to disease severity - a retrospective single-center study" can be published after minor revision - the authors edited the manuscript in sense of reviewers & the quality improved after revion.
- design should be clearer
- conclusion could be extended
Author Response
Dear Reviewer,
thank you for rising the important points. In the following document we have improved/supplemented the listed points.

Reviewer 4 Report
The article now is acceptable to be published.
Author Response
Dear Reviewer,
thank you for your efforts.